# Optimization of the Brewing Process and Analysis of Antioxidant Activity and Flavor of Elderberry Wine

**Huaqiang Cao** [1,2,†]**, Meiyu Bai** [1,2,†]**, Yueyue Lou** [1,2]**, Xiaotian Yang** [3]**, Chenchen Zhao** [1,2]**, Kuan Lu** [1]
**and Pengpai Zhang** [1,2,*]

1 School of Life Sciences, Henan University, Kaifeng 475004, China; ccchq123@163.com (H.C.)
2 Engineering Research Center for Applied Microbiology of Henan Province, Kaifeng 475004, China
3 Laboratory of Cancer Biomarkers and Liquid Biopsy, School of Pharmacy, Henan University, Kaifeng 475004, China
* Correspondence: bio_apai@163.com
† These authors contributed equally to this work.

**Abstract:** Fruit wines have high nutritional value and good palatability. However, fruit wine made from a single fruit type does not have good enough flavor and nutritional quality. Therefore, flavorsome fruit wines made from a variety of fruits should be developed as a matter of urgency. In this study, the raw material of elderberry wine was used to explore the production technology of mixed juice wines; the fruits selected were apple, lychee, pear, blueberry, and elderberry. We utilized a single-factor experiment and the response surface method (RSM) approach to optimize the fermentation procedures; the results show that the solid–liquid ratio was 1:7.5, the amount of yeast inoculation was 0.68 g/L, the fermentation temperature was 20 °C, and the added sugar content was 120 g/L. Under these process conditions, a verification test was carried out in a 35 L fermenter. The results showed that the alcohol content, residual sugar content, total acidity, total phenol content, and total flavonoid content of the elderberry wine were, respectively, 7.73% vol, 8.32 g/L, 9.78 g/L, 8.73 mg/mL, and 1.6 mg/mL. In total, 33 volatile components were identified in the resulting elderberry wine. It achieved a harmonious aroma and fruit flavor, a homogeneous and transparent liquid phase, a pleasant taste, and a sensory evaluation score of 95. The antioxidant activity experiments showed that elderberry had a certain antioxidant capacity, and that fermented elderberries had significantly higher antioxidant ability than unfermented ones.

**Keywords:** elderberry; fruit wine; response surface; optimization; antioxidant

## 1. Introduction

Fruit wine is an alcoholic beverage made of fruits other than grapes, and it may also have flavors drawn from other fruits, plants, and herbs [1]. It contains certain common fruits and alcohol and was the first wine that human beings learned to brew. Fruit wine was made by the Sumerians and ancient Egyptians as early as 6000 years ago. Rural families often brewed fruit wines to drink, such as plum wine, peach wine, kiwifruit wine, and so on [2]. Fruit wine not only has historical precedence but also a new vitality in the current era. From wine made from a single species to the current wines made from a variety of fruit ingredients, the industry is booming. As a result of the fashionable nature of fruit wines and their easy-to-drink characteristics, the novel fruit wine market has been warmly welcomed around the world [3]. Today, with improvements in people's living standards and increasing attention to the importance of health, fruit wine is gradually becoming more popular. However, traditional fruit wine is created by the fermentation of one type of fruit, meaning that the result has a monotonous taste and low nutritional value. Compound fruit wine, a type of fruit wine made by fermenting a variety of fruits, offers the benefits of high nutrition, high quality, and a rich flavor [4]. At present, there are few reports on research into compound fruit wines, so there is an urgent need to develop and perfect a

mature production process and technology to meet people's demand for healthy drinking choices [5].

*Sambucus nigra* L., also known as elderberry, is a deciduous shrub in the honeysuckle family that is found in Europe, North Africa, the United States, and northeast, north, east, and central China [6]. Its roots, stems, leaves, flowers, and fruits have medicinal properties and can be used to treat inflammatory reactions, gout, fractures, and other conditions [7]. The fruit of the elderberry, also known as the elderberry blackberry, is a red to deep purplish black berry that grows in clusters. Berries, which contain a variety of health-enhancing ingredients, such as anthocyanins, fatty acids, vitamins, polyphenols, and flavonoids, have significant antioxidant capacities [8,9]. In the modern diet, they are processed into drinks, jams, fruit juices, condiments, etc., which have a high edible value and are very beneficial in enhancing human health [10]. Recent research has shown that elderberry can also improve immunity to respiratory diseases, help shorten the duration of a cold or flu, and have a significant positive effect on preventing, treating, or curing COVID-19 [11].

Blueberries, pears, apples, and lychees are widely grown in China. These four fruits are consumed fresh and it is difficult to store them, leading to a serious waste of resources [12]. As a result, using them as fermentation raw materials to produce fruit wines, extend the storage time, and achieve a longer consumption period, means that fruit wines are beneficial and can offer huge economic benefits for fruit farmers. Apples, pears, lychees, and blueberries are used as brewing materials. The reason for this is that they each have their own special flavor, taste, color, and fragrance, and these attributes would not yield as rich a fruit wine if just one type of fruit were used to brew it [12]. Elderberry wine is obtained by mixing and fermenting elderberry with several fruits so that it not only retains the taste of the original fruit but also introduces a variety of ingredients with health benefits [8,13]. This practice enriches the variety of fruit wine while also improving the quality [14].

The quality of fruit wine is affected by many factors, such as fermentation temperature, fermentation time, yeast, sugar content, and so on. For example, increasing fermentation temperature tends to increase or reduce the fruit wine's aroma and change the fruit wine's color [15]. Therefore, it is necessary to control the fermentation conditions. The response surface method (RSM), which has three processes for experimental design, predictive response, and result verification, is a mathematical model used to optimize tests. In this way, we can establish the target products to the maximum extent during production [16]. In this study, fruit concentrates of apple, pear, lychee, and blueberry were used as raw materials, and elderberry was added to the base wine to undergo fermentation [17]. To optimize the fermentation results and the fermentation conditions, the RSM and the Box–Behnken design (BBD) were used. At the same time, the physicochemical properties of fermented elderberry wine were determined, then the volatile fragrance was analyzed. This elderberry wine has reference significance for the development of new products.

## 2. Materials and Methods

### 2.1. Materials

Apple juice concentrate, pear juice concentrate, lychee juice concentrate, and blueberry juice concentrate were bought from Zhejiang Dexin Beverage Co., Ltd., Jiaxing County, China. Elderberry extract was purchased from Snott Biotechnology Co., Ltd., Baoji County, China. Sucrose was obtained from the shopping mall in Kaifeng County, China. EC118 active dry yeast was provided from Yantai Mansen Trading Co., Ltd., Yantai County, China. L-ascorbic acid(AP) was obtained from Tianjin Kemiou Chemical Reagent Co., Ltd., Tianjin County, China. DPPH was purchased from TCI Chemical Industrial Development Co., Ltd., Shanghai County, China. ABTS reagent was purchased from Thermo Fisher Scientific (China) Co., Ltd., Shanghai County, China.

## 2.2. Methods

Before the experiment, according to the four kinds of fruit flavor, color, acidity, sugar, taste, and other factors established in the preliminary experiment and combined with the multi-factor analysis, the optimal volume ratio of apple, pear, lychee, and blueberry mixture was 6:2:2:1. We used a measuring cylinder to accurately measure the various fruit juices, mixing them in proportion, then pouring them into a container. After that, a certain amount of water was added to dilute the mixed juice, and the juice was adjusted to a suitable sugar level. The mixture was subsequently diluted with distilled water to yield 5 g/L elderberry extract. The pasteurization technique was used to sterilize the raw fermentation material, which was then added to a 500 mL glass fermentation bottle. For yeast activation, the dry yeast was activated for 15–30 min with more than 5 times the amount of distilled water at about 35 °C. After the addition of the activated yeast at a suitable ratio and adjustment of the temperature to the proper level, fermentation started. The main fermentation stage is about 18 days, and the post-fermentation stage is conducted at 3 °C (day 20). Then, the alcohol content, total residual sugar, total acid content, sensory evaluation, and other related indices were determined and analyzed.

### 2.2.1. Single Factor Experiments

A preliminary investigation of the factors affecting the taste and quality of fermented elderberry wine was conducted using single-factor experiments on 4 factors, including the solid–liquid ratio, sugar sweetening, yeast addition, and fermentation temperature. The factors chosen were the different solid–liquid ratios (1:5, 1:6, 1:7, 1:8, and 1:9), sugar sweetening (100, 110, 120, 130, and 140 g/L), yeast addition (0.5, 0.6, 0.7, 0.8, and 0.9 g/L) and fermentation temperature (15, 18, 21, 24, and 27 °C). Experiments were carried out using the control variable method. Alcohol and sensory scores were used as measurement indices, and Origin8.5 was used for mapping.

### 2.2.2. Experimental Design

A Box–Behnken design (BBD) was used to optimize fermentation, with 3 factors and 3 levels for each variable. Based on the results of the single-factor experiment, a response surface experiment was chosen in order to obtain the optimal fermentation conditions for elderberry wine. The independent variables that had the greatest influence on experimental design were the solid–liquid ratios (1:6, 1:7, and 1:8), yeast addition (0.6, 0.7, and 0.8 g/L), and fermentation temperature (18, 21, and 24 °C). BBD was used to evaluate the combined effect of the 3 independent variables. This consisted of 15 experimental runs, of which the central point (3 replicates) was used for the optimization of the fermentation conditions. The ranges of the independent variables in the design were prescribed into three levels, coded −1, 0, and +1, as depicted in Table 1. The dependent variables that were measured were the sensory score (Y1) and alcohol content (Y2). The levels of independent variables and the design matrix are shown in Table 2. All the experimental data obtained from the designed experiments are fitted with a quadratic model using Equation (1):

$$Y = \beta_0 + \beta_1 A + \beta_2 B + \beta_3 C + \beta_{11} A^2 + \beta_{22} B^2 + \beta_{33} C^2 + \beta_{12} AB + \beta_{13} AC + \beta_{23} BC \quad (1)$$

where Y is the predicted response and A, B, and C correspond to the independent variables for solid-liquid ratio, yeast inoculum, and fermentation temperature. $\beta_0$ represents the intercept; $\beta_1$, $\beta_2$, and $\beta_3$ are the linear coefficients; $\beta_{12}$, $\beta_{13}$, and $\beta_{23}$ are the interaction coefficients; $\beta_{11}$, $\beta_{22}$, and $\beta_{33}$ are the quadratic coefficients; Y is the response value.

**Table 1.** The coded and actual values of factors in the BBD.

| Factor | Name | Low Actual | High Actual | Low Coded | High Coded |
|---|---|---|---|---|---|
| A | Solid–liquid ratio | 1:8 | 1:6 | −1 | 1 |
| B | Yeast inoculum | 0.6 | 0.8 | −1 | 1 |
| C | Fermentation temperature | 18 | 24 | −1 | 1 |
| Response | Name | Observed | Min | Max | Mean |
| Y1 | Sensory score | 15 | 60 | 96 | 78 |
| Y2 | Alcohol content | 15 | 6.86 | 8.66 | 7.76 |

Note: (In the RSM, the Solid-liquid ratio: 1:8 is regarded as 8, 1:7 as 7, and 1:6 as 6).

**Table 2.** The BBD matrix and responses.

| Run | Independent Variable | | | Response | |
|---|---|---|---|---|---|
| | A | B | C | Sensory Score (Y1) | Alcohol Content/% Vol (Y2) |
| 1 | −1 | −1 | 0 | 80 | 8.09 |
| 2 | 1 | 1 | 0 | 71 | 7.72 |
| 3 | 0 | 0 | 0 | 94 | 8.35 |
| 4 | 1 | 0 | −1 | 87 | 6.86 |
| 5 | 1 | 0 | 1 | 85 | 7.29 |
| 6 | 0 | −1 | −1 | 86 | 7.62 |
| 7 | 0 | 0 | 0 | 96 | 8.32 |
| 8 | 0 | 1 | −1 | 67 | 7.52 |
| 9 | 0 | 1 | 1 | 65 | 8.66 |
| 10 | 0 | 0 | 0 | 93 | 8.30 |
| 11 | 0 | −1 | 1 | 69 | 7.74 |
| 12 | −1 | 1 | 0 | 60 | 8.29 |
| 13 | 1 | −1 | 0 | 76 | 6.94 |
| 14 | −1 | 0 | 1 | 73 | 8.43 |
| 15 | −1 | 0 | −1 | 80 | 7.83 |

Note: Each experiment was carried out in triplicate.

### 2.2.3. Elderberry Wine Index Determination

The contents of alcohol, total residual sugar, and total acid were determined according to the standard GB/T 15038–2006 "Analytical methods for wine and fruit wine". After referring to the literature [18], the total polyphenol and total flavonoid contents were also determined.

### 2.3. Sensory Evaluation

According to the literature review [19], ten members of the sensory evaluation panel, five men and five women, were trained in the analytical methods necessary for elderberry wine. They evaluated the elderberry wine for the following properties: aroma (0–20 points), color (0–20 points), clarity (0–20 points), taste (0–20 points), and typicality (0–20 points). The sensory scale, scores, and detailed rules are shown in Table 3.

### 2.4. Analysis of Volatile Flavor Substances

Compared to previously published research [20], our method was slightly modified. First, 100 mL of the elderberry wine sample was placed in a 250 mL separator funnel. An equal volume of dichloromethane was added. The hybrid liquid was shaken for 5 min. After standing and delaminating, the extraction solutions were released from the separator funnel and were further subjected to anhydrous sodium sulfate to remove excess water. The extraction solutions were concentrated down to 5 mL. The extracted samples, diluted with dichloromethane to a suitable concentration, were filtered with a 0.22 mm microporous filter membrane and analyzed using GC-MS (Gas Chromatography-Mass Spectrometer).

**Table 3.** Sensory scoring criteria.

| Index | Description | Score |
|---|---|---|
| aroma (20) | The aroma is fragrant and clear, with obvious fruit ester aroma, fragrance coordination, and odorless | 15–20 |
| | The flavor is plain, the fragrance is not coordinated and odorless | 10–15 |
| | No fruity aroma, unpleasant odor | 0–10 |
| color (20) | Ruby red, uniform color distribution | 15–20 |
| | Pink, uneven color distribution | 10–15 |
| | Scarlet, uneven color distribution | 0–10 |
| clarity (20) | Clear and bright, inclusion-free | 15–20 |
| | Slight loss of light, no obvious impurities | 10–15 |
| | Turbid, with obviously suspended solids | 0–10 |
| taste (20) | Fresh and delicate taste, sweet and sour, tastes delicious and special | 15–20 |
| | Refreshing taste, sour or sweet | 10–15 |
| | No obvious taste, too sour or too sweet | 0–10 |
| typicality (20) | Obvious fruit flavor and characteristics | 15–20 |
| | Part of the fruit flavor and characteristics | 10–15 |
| | Almost nothing of the fruit flavor and characteristics | 0–10 |

The GC (Gas Chromatography) conditions were as follows: separation was carried out on a capillary column (HP-5MS 30 m; 0.25 mm ID; 0.25 μm thickness). The injection port was maintained at 240 °C, with an injection volume of 1 μL. The flow rate was 1.8 mL/min and the diversion ratio was 2:1. The following temperature program was used: 3 min at 35 °C, then an increase to 115 °C at 10 °C/min, then an increase to 235 °C at 10 °C/min and, finally, was held there for 10 min. The MS (Mass Spectrometer) conditions were as follows: the electron energy was 70 eV, and the ion source temperature was 200 °C.

The results were analyzed using the NIST (The National Institute of Standards and Technology). Chromatographic peaks were identified by comparing the mass ions of each peak with the NIST MS Library. Quantitative analysis was performed using the area normalization method.

*2.5. Antioxidant Activity Experiment*

The sample preparation was as follows: sample 1 held the elderberry extract solution (5 g/L); sample 2 held the elderberry wine without elderberry extract; sample 3 held the elderberry wine. Antioxidant activity was analyzed through the DPPH, ABTS$^+$, and (O$_2^-$) assays according to a previous study [21], with some modifications.

2.5.1. Determination of (DPPH) Scavenging Rate

In a 96-well plate, a properly prepared sample solution was mixed with a DPPH solution. Taking measurements at 517 nm, following 30 min of reaction at room temperature, the absorbance (A) was determined. V$_C$ was used as the positive control. The absorbance of the mixture containing the sample and DPPH solution is A$_1$. The absorbance of the sample blended with ethanol is A$_2$, and the absorbance of ethanol mingled with DPPH solution is A$_0$. The scavenging rate of the compound was calculated using the following formula in Equation (2):

$$\text{DPPH Scavenging Rate (\%)} = \frac{A_0 - (A_1 - A_2)}{A_0} \times 100\%. \tag{2}$$

2.5.2. Determination of (ABTS$^+$) Scavenging Rate

Each sample was mixed with ABTS solution in a 96-well clear plate and reacted for 5 min at room temperature, then the absorbance of samples at 734 nm was measured. V$_C$ was used as the positive control. The absorbance of the sample mixture and ABTS working

solution is $A_1$; the sample was then replaced with ethanol as the absorbance control, $A_0$. The scavenging rate of the compound was calculated using the following formula, Equation (3):

$$\text{ABTS Scavenging Rate } (\%) = \frac{A_0 - A_1}{A_0} \times 100\%. \tag{3}$$

### 2.5.3. Determination of the Superoxide Anion Radical ($O_2^-$) Scavenging Rate

Each sample was mixed with Tris-HCl buffer and pyrogallol ($A_1$) in a 96-well clear plate and reacted for 6 min at 37 °C. Next, the reaction was terminated by the addition of 0.1 mol/L HCl and the absorbance of the samples at 320 nm was measured. $V_C$ was used as the positive control and $H_2O$ was used as the reference liquid ($A_0$). Pyrogallol was replaced with $H_2O$ as the absorbance control ($A_2$). The scavenging rate of the compound was calculated using the following formula, Equation (4):

$$O_2^- \text{ Scavenging Rate } (\%) = \frac{A_0 - (A_1 - A_2)}{A_0} \times 100\% \tag{4}$$

## 3. Results and Analysis

### 3.1. Optimal Fermentation Conditions for Elderberry Wine

#### 3.1.1. Single-Factor Experiment Results

(a)    Influence of the solid–liquid ratio on fermentation

As a fermentation raw material, concentrated fruit juice was used instead of fresh fruits. The use of concentrated fruit juice instead of fruit as the fermentation raw material has the advantages of long service life, repeated use, easy operation, low cost, and better reproducibility. Concentrated fruit juice contains a high sugar content, high levels of acid, and other organic components. Therefore, the concentrate needs to be diluted with water before fermentation. The solid–liquid ratio was studied in the range of 1:5–1:9, and the effects on the sensory scores and alcohol content are shown in Figure 1a. With the extended solid–liquid ratio, the alcohol content decreased gradually, and the sensory score first increased and then decreased. When the solid–liquid ratio was 1:7, the sensory score reached a maximum of 94, and the alcohol content was 7.82 vol. The flavor was now sweet and sour, delicious, and unique. Regardless of whether the solid–liquid ratio was greater than or less than 1:7, the wine had a heavy body, was not pleasant, and had a bad taste. The elderberry wine flavor was also affected. The optimal condition is thus a solid–liquid ratio at 1:7. Therefore, in this study, a solid-liquid ratio at 1:7 was indicated as appropriate for subsequent experiments.

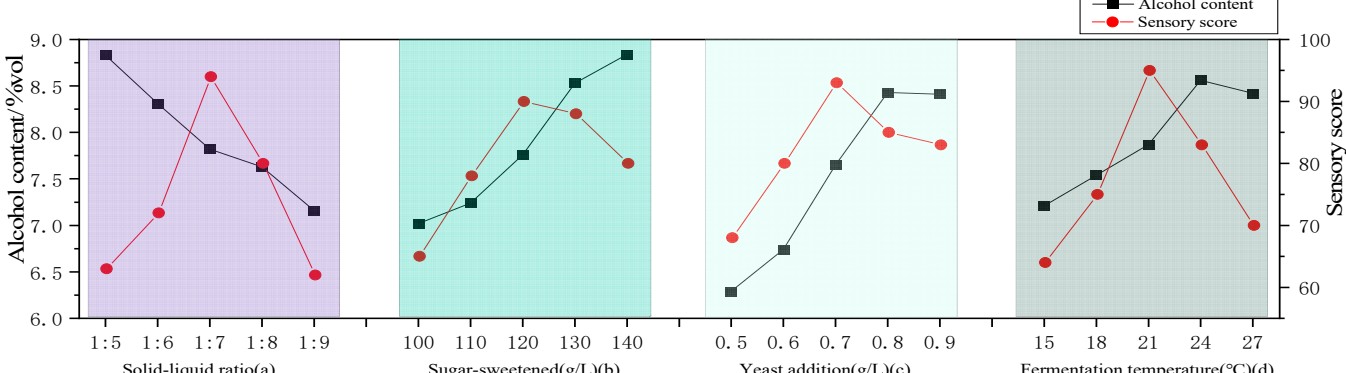

**Figure 1.** The results of the single-factor test.

(b)    Effect of sugar-sweetening on fermentation

Generally, the average sugar content of ripe fruits is about 100 g/L, which is not high enough for the fermentation process to produce enough alcohol. For yeasts to produce alcohol, additional sucrose must be added. Through fermentation, the changes in sweetness

on the sensory score and alcohol content are shown in Figure 1b. The trends can be discerned in Figure 1b. With each increase in sugar-sweetening, the alcohol content gradually increased, and the sensory score first increased and then gradually decreased. However, the addition of sucrose resulted in a large amount of alcohol, which would cause intensity rather than mellowness and disrupt the balance of the wine. Therefore, the conditions of 120 g/L were selected as the best conditions. At this point, the alcohol content was 7.76% vol, and the sensory score was 90. The resulting elderberry wine has a distinctive, harmonious, and mellow taste, along with a fruit flavor, suggesting coordination between olfaction and taste.

(c)    Effect of yeast inoculum on fermentation

Yeast is the driving force of fermentation and has a great influence on the flavor and quality of elderberry wine. The effect of yeast inoculum on fermentation can be seen in Figure 1c. As the yeast dosage increased, the alcohol content also increased, then finally tended to balance. The sensory score gradually increased first and then decreased. This indicated that the yeast was added in very small amounts so that the sugar in the juice did not trigger full fermentation. Conversely, by adding too much yeast, off-flavor compounds derived from the yeast can be produced during fermentation. Based on this finding, 0.7 g/L was selected as the optimal yeast inoculum. Under these conditions, the alcohol content was 7.65% vol, and the sensory score was 93. Fermented elderberry wine has a strong fruit flavor and a coordinated body. The dynamics of the flavors are clear.

(d)    Effect of temperature on fermentation

Temperature is an important factor affecting elderberry wine production throughout the fermentation process. For instance, high temperatures will produce methanol, diacetyl, and other impurities, while low temperatures will lead to incomplete fermentation. The influence of fermentation temperature on the sensory score and the alcohol content is shown in Figure 1d. With an increase in temperature, the alcohol content of elderberry wine first increased and then decreased. The sensory score also first increased and then decreased. When the fermentation temperature was lower than 21 °C, the inhibitory effect is enhanced and the fermentation process that generates the aroma substances was impaired. The fermentation process was slowed down, and higher concentrations of residual sugars were observed. When the temperature was higher than 21 °C, the fermentation process was even more exergonic, with a strong fusel and lipid taste. Therefore, 21 °C was the best reaction temperature. The alcohol content of elderberry wine was 7.87% vol at this temperature, and the sensory score was 95. The resulting elderberry wine has an elegant body and a balanced composition; there were more fragrant ingredients and the alcohol and aroma components in the elderberry wines were relatively more harmonious.

3.1.2. Fitting the Response Surface Models

The RSM experiments [22] are designed to optimize interested responses that are affected by several variables. The influence of the solid–liquid ratio, yeast inoculum, and fermentation temperature on the sensory score and alcohol content were investigated using RSM. The sensory score and alcohol content were chosen as the response values. To examine the statistical significance of factors in the model, an analysis of variance (ANOVA) and regression analysis were conducted; the results are shown in Table 4.

The role of each variable and their second-order interactions is explained by a second-order polynomial model in coded units, based on multiple regression analysis using experimental data. The developed models are shown in Equations (5) and (6).

$$\text{Sensory score}: Y1 = 94.33 + 3.25A - 6.00B - 3.50C + 3.75AB + 1.25AC + 3.75BC \\ -6.54A^2 - 16.04B^2 - 6.54C^2 \tag{5}$$

$$\text{Alcohol content}: Y2 = 8.32 + 0.48A + 0.22B + 0.29C - 0.14AB + 0.042AC + 0.26BC \\ -0.42A^2 - 0.14B^2 - 0.30C^2 \tag{6}$$

A linear regression model was constructed and the coefficient of determination, $R^2$, and F value were found. Based on Table 4, the overall results of the performance parameters are shown as listed below: the $R^2$ for the responses of both sensory score and alcohol content were 0.9799 and 0.9929. The F values of both regression models were 27.02 and 77.49, and the *p*-values of both regression models were 0.001 and 0.0001 < 0.05; the *p*-values of both where the lack of fit was non-significant were 0.1897 and 0.0625 > 0.05, which shows that model is significant. This also suggests that the prediction model can be tested against the mathematical model in the equation.

**Table 4.** The ANOVA evaluation of linear, interaction, and quadratic terms for sensory score and alcohol content response variables and the coefficients of the model prediction.

| Source | SS | DF | MS | F Value | *p*-Value | SS | DF | MS | F Value | *p*-Value |
|---|---|---|---|---|---|---|---|---|---|---|
| | Sensory Score ($Y_1$) (a) | | | | | Alcohol Content ($Y_2$) (b) | | | | |
| Model | 1734.73 | 9 | 192.75 | 27.02 | 0.001 | 4.20 | 9 | 0.47 | 77.49 | 0.0001 |
| A | 84.50 | 1 | 84.50 | 11.85 | 0.0184 | 1.83 | 1 | 1.83 | 304.67 | <0.0001 |
| B | 288.00 | 1 | 288.00 | 40.37 | 0.0014 | 0.4 | 1 | 0.40 | 67.29 | <0.0001 |
| C | 98.00 | 1 | 98.00 | 13.74 | 0.0139 | 0.66 | 1 | 0.66 | 108.92 | 0.0004 |
| AB | 56.25 | 1 | 56.25 | 7.89 | 0.0376 | 0.084 | 1 | 0.084 | 13.97 | 0.0001 |
| AC | 6.25 | 1 | 6.25 | 0.88 | 0.3922 | 0.0072 | 1 | 0.00723 | 1.20 | 0.0135 |
| BC | 56.25 | 1 | 56.25 | 7.89 | 0.0376 | 0.26 | 1 | 0.26 | 43.22 | 0.3232 |
| $A^2$ | 158.01 | 1 | 158.01 | 22.15 | 0.0053 | 0.66 | 1 | 0.66 | 109.73 | 0.0012 |
| $B^2$ | 950.16 | 1 | 950.16 | 133.2 | <0.0001 | 0.073 | 1 | 0.073 | 12.10 | 0.0001 |
| $C^2$ | 158.01 | 1 | 158.01 | 22.15 | 0.0053 | 0.33 | 1 | 0.33 | 54.45 | 0.0177 |
| Residual | 35.67 | 5 | 7.13 | | | 0.030 | 5 | 0.006 | | |
| Lack of fit | 31.00 | 3 | 10.33 | 4.43 | 0.1897 | 0.029 | 3 | 0.0096 | 15.17 | 0.0625 |
| Pure error | 4.67 | 2 | 2.33 | | | 0.00127 | 2 | 0.0006 | | |
| total | 1770.40 | 14 | | | | 4.23 | 14 | | | |
| $R^2$ | | | 0.9799 | | | | | 0.9929 | | |

Note: Where SS, DF, and MS stand for sum of squares, degree of freedom, and mean square, respectively. (a) The ANOVA results of the quadratic response surface model for sensory score. (b) The ANOVA results of the quadratic response surface model for alcohol content.

(a)    RSM model for sensory score

Fermentation is a complicated and multi-step process that requires the coordination of many factors, including the physical and chemical factors of the yeast, sugar, acidity, temperature, etc. [23]. These different factors can affect the results of brewing studies. The significance of the F and *p*-values in the linear, interaction, and quadratic terms of the solid–liquid ratio, fermentation temperature, and yeast inoculum are shown in Table 4a. The optimized simulated results revealed the interaction of two out of the three major factors on the sensory score when the third one was fixed at a certain level in the 3D curved surface and 2D contour plots (Figure 2). The model terms A, B, C, AB, BC, $A^2$, $B^2$ and $C^2$ were significant ($p < 0.05$) and were positively affected. The other terms, AC, were inessential ($p > 0.05$), and were negatively affected.

It can be seen that the three factors tested were highly influential in affecting the fraction of the taste and mouthfeel of elderberry wine. By analyzing the effect of the main factors and the F value of the main factors, it can be concluded that the effect of these factors was yeast inoculum > fermentation temperature > solid–liquid ratio. Moreover, the quadratic response 3D surface plot in Figure 2 clearly illustrates the optimization model of solid–liquid ratio, fermentation temperature, and yeast inoculum for the sensory score. Fruit type, fermentation techniques, and fermentation conditions are the main determining factors in the complexity of the wine's sensory characteristics [24]. As showcased in Figure 2A–C, with the increase in inoculum amount and fermentation temperature, the sensory score first decreased and then increased, with noticeable changes. Fermentation temperature influences the aroma profiles and concentration of elderberry wine. Juice fermented at different temperatures produces different aroma profiles. Since then, during

fermentation, increasing the temperature may improve the yeast growth rate, fermentation rate, and cell viability, which results in bad odor and the production of fusel alcohol. In contrast, the temperature decreased, which affected the growth and fermentation rate of yeast and influenced the constitution of fermentation products. The effects of yeast inoculum and fermentation temperature on the measurements are alike. When too much yeast was added, the fermentation rate was excessively promoted and would have negative effects on the flavor and taste of the spirits. In turn, fermentation would be inhibited, resulting in the wine not maturing after fermentation. In Figure 2a–c, from the two-dimensional contour, the oval and circular contours reveal that the interaction between the two factors was significant or not significant, respectively [25,26]. As an interaction of the solid–liquid ratio and fermentation temperature, fermentation temperature and the yeast inoculum were significant, while the interaction of fermentation temperature and the solid–liquid ratio was not significant. Sensory score predictive values were 95.12, 95.58, and 95.14.

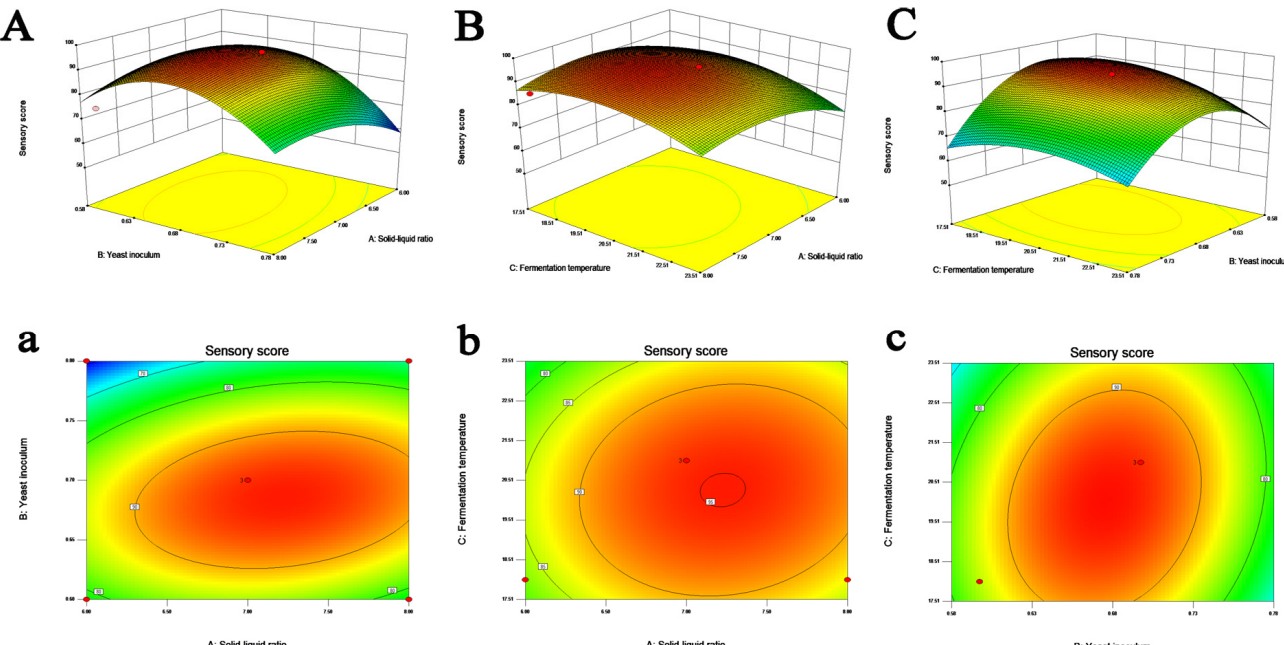

**Figure 2.** Response surface and contour plots for sensory score. (**A**,**a**) Effect of solid–liquid ratio and yeast inoculum. (**B**,**b**) Effect of solid–liquid ratio and fermentation temperature. (**C**,**c**) Effect of yeast inoculum and fermentation temperature).

(b)    RSM Model for Alcohol Content

Alcohol, which has been identified as a volatile organic compound, is an important marker of wine. The alcohol had a significant effect on the wine quality parameters. In addition, the production of alcohol is also dependent on various fermentation factors. However, in terms of taste, a higher alcohol content is not better, nor is a lower content better. It has been reported that the appropriate alcohol concentration can contribute to the mellowness of alcoholic beverages [27].

In the context of fitting the model, the alcohol content response surface methodology (RSM) results are shown in Table 4b. From the regression model ($Y_2$) of ethanol concentrations, the value of the determination coefficient ($R^2 = 0.9714$) indicates that only 0.71% of the total variations were not explained by the model. Working from Table 4, the 3D views of interaction effects and the 2D contour plots in the alcohol content are displayed in Figure 3. Among the model terms, A, B, C, AB, AC, $A^2$, $B^2$, and $C^2$ were significant ($p < 0.05$). The interaction of BC, however, had no significant influence on ethanol production in elderberry wine.

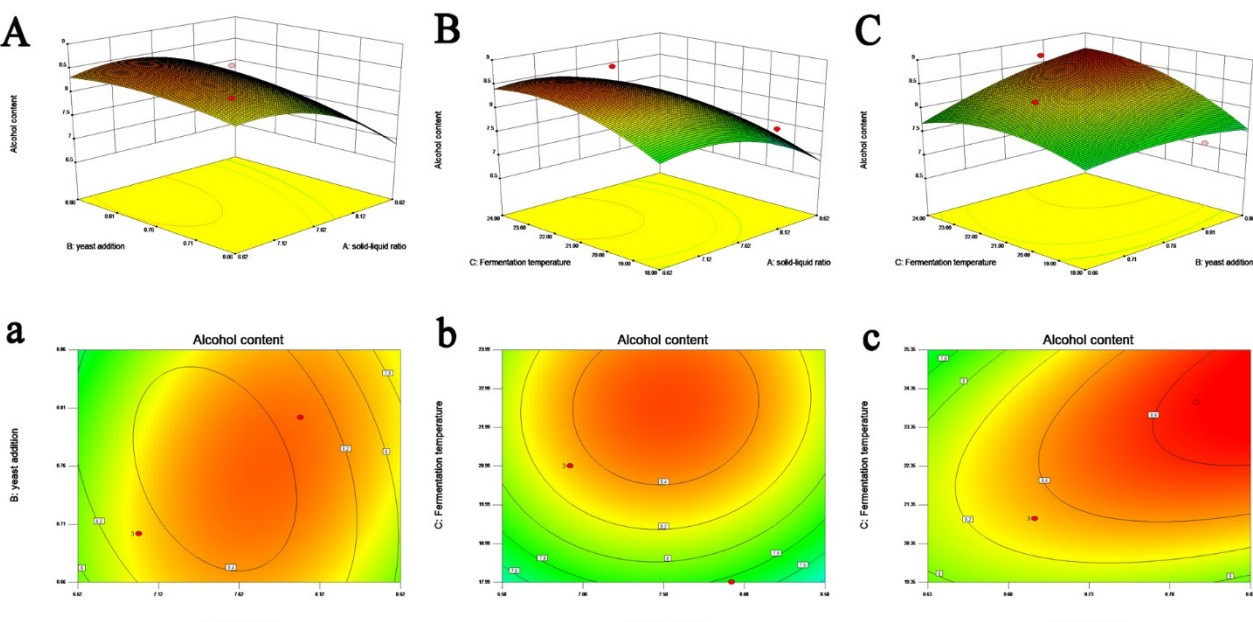

**Figure 3.** Response surface and contour plots for alcohol content. (**A**,**a**) Effect of solid–liquid ratio and yeast inoculum. (**B**,**b**) Effect of solid–liquid ratio and fermentation temperature. (**C**,**c**) Effect of yeast inoculum and fermentation temperature).

From Figure 3A–C, it can be seen that the alcohol content increased rapidly with the initial increase in the concentration of yeast inoculum and fermentation temperature. The reasons for this could be as follows: yeast cell reproduction is temperature-dependent. Higher suitable temperatures could increase the suitable period in which to produce yeast, which could, in turn, increase the number of yeast particles dispersing. When the temperature is gradually decreased, the growth of yeast is strongly inhibited, thereby reducing fermentation efficiency. Or, if the yeast inoculum is low, the yield of the alcohol will also be reduced. In parallel, a high solid–liquid ratio means that a high sugar content is much preferred in alcohol fermentation, which may raise the content of ethanol. Hence, the metabolic reaction of the fermentation process is progressively decreased as the solid–liquid ratio decreases, as can be seen in Figure 3a–c [28]. The contour line is oval, indicating that the interaction of any two of the three factors is significant. We can obtain the optimum predicted value from the smallest ellipse in the contour diagram. The result is that the predicted value of the interaction effect of the solid–liquid ratio and yeast inoculum is 8.499, the interaction effect of the solid-liquid ratio and fermentation temperature is 8.5341, and the interaction effect on fermentation temperature and yeast inoculum is 8.7435. It is clear that most predicted points are close to the experimental values.

### 3.1.3. Validation and Verification of the Optimized Conditions

The response surfaces (Figures 2 and 3) obtained were used as a basis for optimizing the investigated parameters for the fermentation of elderberry wine production. The resulting response surfaces showed the effect of the solid–liquid ratio, fermentation temperature, and yeast inoculum on the ethanol and sensory scores. According to our previous experiments, the optimization should be aimed at a high score and moderate alcohol level. Because of its negative sensory properties, such as the suppression of fruity notes, alcohol is a limiting factor in this study [29]. Therefore, we considered prediction using each of these models individually. The sensory score's predictive value was set to maximum, but the predictive value of the alcohol content range was set from 6.86 to 7.66. As a result, the optimized values obtained from the model for the solid–liquid ratio, yeast inoculum, and fermentation temperature were 1:7.57, 0.68 g/L, and 19.49 °C. Under these conditions, the highest sensory score possible is 94.30, and the alcohol has a volume of 7.66%.

The fermentation conditions were adjusted to a solid–liquid ratio of 1:7.5, a yeast inoculum of 0.68 g/L, and a fermentation temperature of 20 °C, based on the results of the response surface optimization experiment and the real-world situation. Verification tests can also be carried out under optimized fermentation conditions in a 35 L fermenter. The physical and chemical index is shown in Table 5.

**Table 5.** Physical and chemical index.

| | Alcohol Content /% Vol | Residual Sugar /(g/L) | Total Acid/(g/L) | Total Phenol /(mg/mL) | Total Flavone /(mg/mL) |
|---|---|---|---|---|---|
| Pre-fermentation | — | 25.68 ± 0.96 | 9.72 ± 0.23 | 3.34 ± 0.09 | 0.7 ± 0.06 |
| Post-fermentation | 7.73 ± 0.13 | 8.32 ± 0.82 | 9.78 ± 0.19 | 8.73 ± 0.11 | 1.6 ± 0.11 |

Note: The experiment was carried out in triplicate, and all data were expressed as means ± standard deviation (SD).

Table 5 shows that the alcohol content of the elderberry wine is 7.73%, which is close to the predicted value. The residual sugar concentrations are 8.32 g/L, the total acid content is 9.78 g/L, the total phenol content is 8.73 mg/mL, which is up from 5.39 mg/mL compared with pre-fermentation, and the total flavonoids are 1.6 mg/mL, which is 0.9 mg/mL higher than at pre-fermentation. Compared with the index of pre-fermentation and post-fermentation, the contents of total phenol and total flavone were significantly increased, indicating that the active substance is more readily released from the elderberry. These results imply that fermentation caused the deglycosylation of phenolic glycosides in elderberries [23,30]. Enzymatic hydrolysis of biomass is a critical step in fermentation that degrades total phenolics and flavonoids into simple small molecules, which is important to improve the quality of elderberry wine [31].

From Figure 4, it can be seen that in the early stages of fermentation, the mixed juice presented a good odor. With the extension of fermentation time, there was a clear change in parameters for every single sensory characteristic, which showed that: the fruit fragrance was prominent and harmonious; the color was ruby red; the clarity was clear and bright with no impurities; the mouthfeel was fresh and delicate, with a sweet and sour flavor and a satisfying aftertaste. It had obvious typicality, with a significantly fruity character and fruity flavor. The final sensory score was 95, closing in on the theoretically predicted value. The samples of elderberry juice were rich in D-terpendiene, and the substance had a flower-like aroma. Subsequently, the juice was fermented and regenerated into damustrone, which contains floral, fruity, and lilac aroma characteristics [32].

### 3.2. Analysis Results for Flavor Substances

Flavor substances are the heart and soul of fruit wine and are critical indicators of its quality. Product quality can be predicted at an early stage of the manufacturing process by qualitative analysis of the flavor substances in fruit wine, which can direct us to optimize the reaction conditions. The main aromatic active compounds in fruit wine are aldehydes, esters, and alcohols [33].

According to the results from GC-MS and NIST software analysis, the total ion flow chromatogram of elderberry wine is shown in Figure 5; 33 volatile chemical compounds were identified, with the contents shown in Table 6.

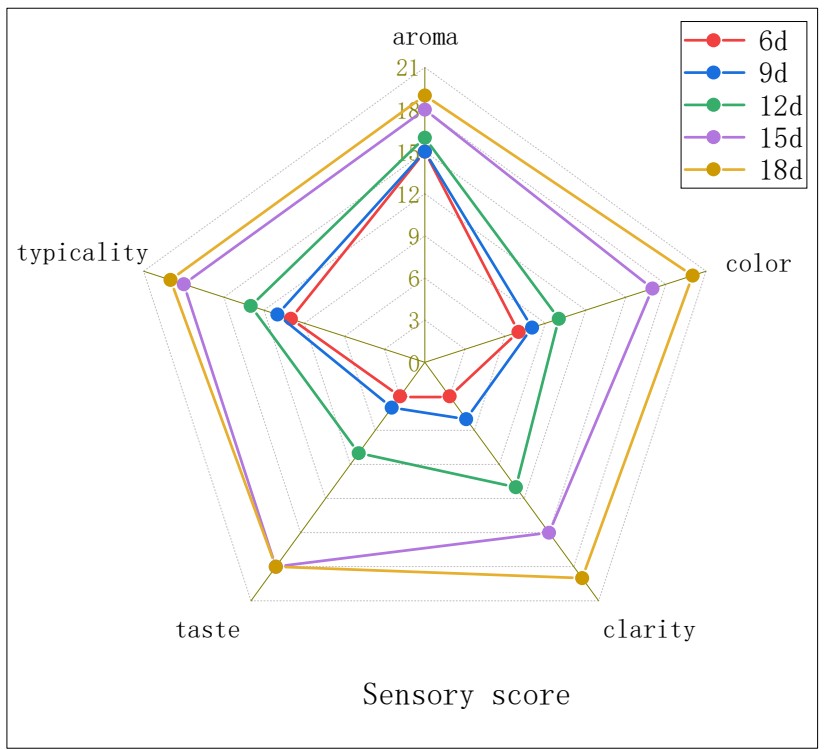

**Figure 4.** Sensory score changes according to fermentation time.

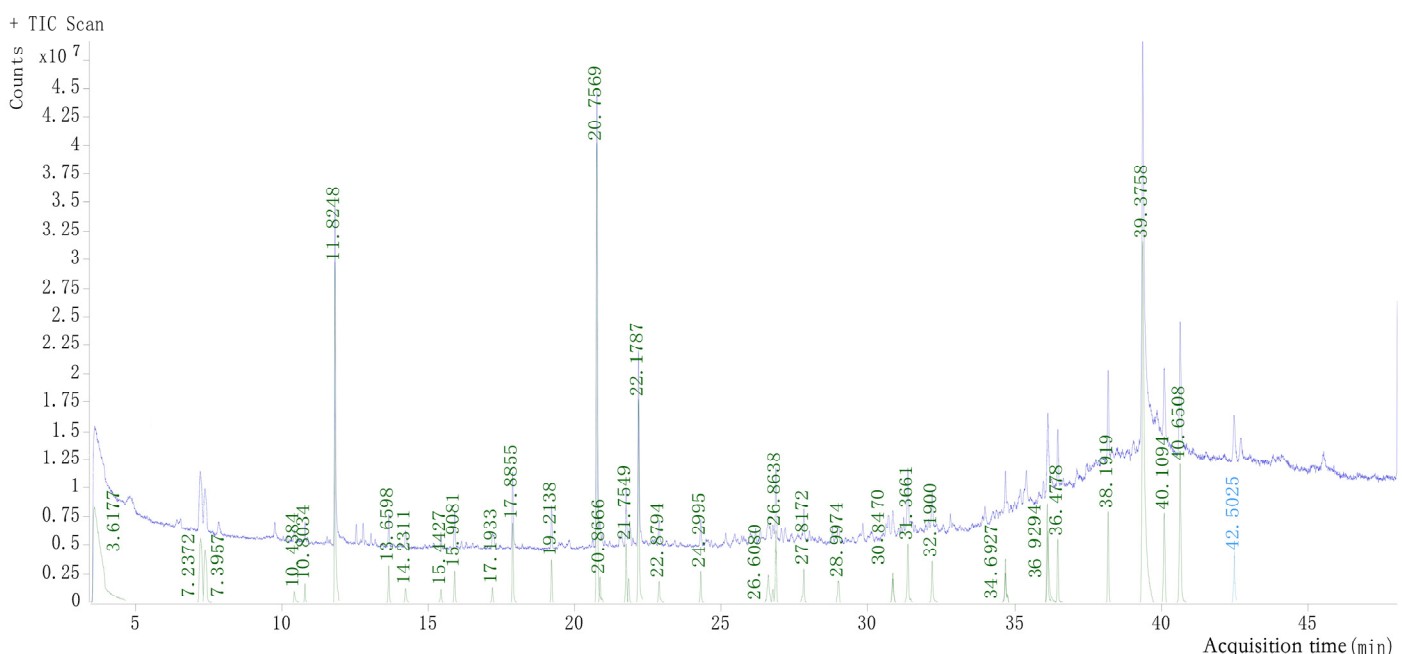

**Figure 5.** Total ion chromatogram of the aroma compounds in elderberry wine.

**Table 6.** Aroma components in elderberry wine.

| No. | Compounds | Descriptors | Relative Peak Area/% |
|---|---|---|---|
| 1 | Ethyl caprylate | Lychee | 19.183 |
| 2 | Ethyl butyrate | Fruity, Apple, | 11.779 |
| 3 | Ethyl caprate | Grape, Fruity | 2.735 |
| 4 | 2-methyl pentane-3-acetate | Fruity | 0.322 |
| 5 | Ethyl acetate | Fruity | 0.808 |
| 6 | Ethyl 4-ketovalerate | Floral | 0.421 |
| 7 | Triglyceride triacetate | Fruit, fruity, sweet | 0.312 |
| 8 | Methyl benzoate | Holly oil, grass | 0.720 |
| 9 | Glycerol 1,2-diacetate | — | 1.870 |
| 10 | Methyl dimethoxy acetate | Fruity | 0.949 |
| 11 | Ethyl 2-methyl butyrate | Apple, Strawberry, grape | 3.590 |
| 12 | Diethyl succinate | Pleasant scent | 0.723 |
| 13 | Ethyl 3-methylbutyrate | Blueberry, fruity, floral | 2.815 |
| 14 | Isoamyl acetate | Banana | 2.168 |
| 15 | Methyl acetate | Apples, peaches | 16.123 |
| 16 | Ethyl benzoate | Fruity, floral | 2.818 |
| 17 | Ethyl lactate | Fruity, ester, grass | 4.748 |
| 18 | Benzyl alcohol | Mint, sweet, fruity | 0.311 |
| 19 | 1-Hexanol | Floral, sweet | 1.422 |
| 20 | 3-Methyl-1-butanol | Sweet | 0.602 |
| 21 | 2-Ethyl hexanol | Flowers, grass | 0.721 |
| 22 | 2-Methyl-1-propanol | Mellow, grassy | 1.770 |
| 23 | Cis-3-hexene-1-ol | Sweet, grass | 1.325 |
| 24 | 2-Nonanone | Honey | 0.343 |
| 25 | Alpha-Ionone | Fruity | 1.161 |
| 26 | Beta-damascenone | Rose | 0.942 |
| 27 | 2, 3-butanol ketone | Sweet, fruity | 0.930 |
| 28 | Trans-3-pentene-2-one | Alcohol, sweet | 1.215 |
| 29 | Trans-3-hexenoic acid | Honey, fruity | 7.474 |
| 30 | 2,4-Di-tert-butylphenol | Lemon | 5.209 |
| 31 | N-butyric acid | Fruity | 1.760 |
| 32 | Nonanal | Sweet | 1.540 |
| 33 | N-caprylic acid | Fruity, pineapple, honey | 1.193 |

A total of 33 volatile compounds were identified, including 17 esters, 6 alcohols, 2 acids, and 8 other substances. Alcohols, esters, and terpenoids were predicted to be the main contributors to the fragrance ingredients in elderberry wine. In fact, ethyl caprylate, isoamyl acetate, ethyl caprate, ethyl acetate, N-caprylic acid, and other esters and aldehydes are the main flavoring substances. Among these components, it was found that the largest proportion of ethyl caprylate (19.183%) was in the relative peak area, followed by methyl acetate (16.123%) and ethyl butyrate (11.779%). All these compounds contributed to the characteristic cherry wine aroma, with fruity, lychee, apple, grape, green, and sweet notes [34]. These flavor substances are transformed from fruit scents through fermentation via biological oxidation and biological metabolism. Furthermore, we discovered an active ingredient known as 2,4-di-tert-butylphenol in all the ingredients. This substance had significant antioxidant activity, which may come from the elderberry or may be produced by fermentation [35].

### 3.3. Experimental Results and Analysis of Antioxidant Activity

3.3.1. DPPH Free Radical Scavenging Ability

Elderberry is the berry of the black elder or *Sambucus nigra*, which contains a large number of antioxidant substances, including flavonoids, polyphenols, saponins, and other substances [36]. Therefore elderberry wine has a positive effect on antioxidant activity [37].

The effects of elderberry wine on antioxidant capacity are shown in Figure 6. Following the addition of the sample, the scavenging ability of DPPH free radicals increased steadily. When sample 1 was 60 µL, the scavenging rate of DPPH radicals reached its maximum of 63.55%. When the sample 2 and sample 3 sizes were 80 µL, the scavenging rates of DPPH free radicals reached the maximum, which was recorded at 72.78% and 84.32%, respectively. The results indicated that elderberry and fruit juice had a similar effect on the scavenging of DPPH radicals. Through fermentation, the scavenging rate of DPPH free radicals in elderberry wine was enhanced, which may be related to the degradation of elderberry in the fermentation process and the release of more antioxidant substances [38].

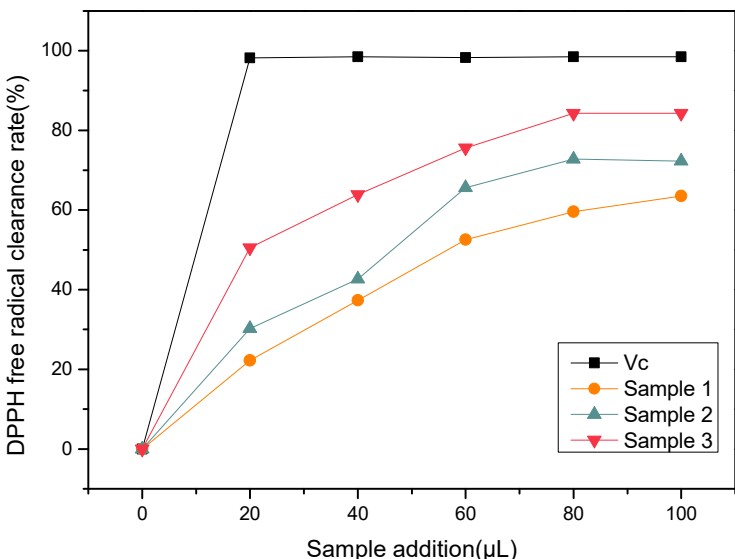

**Figure 6.** DPPH free radical clearance ability.

### 3.3.2. ABTS Free Radical Scavenging Ability

The scavenging ability of ABTS$^+$ cation radicals is shown in Figure 7.

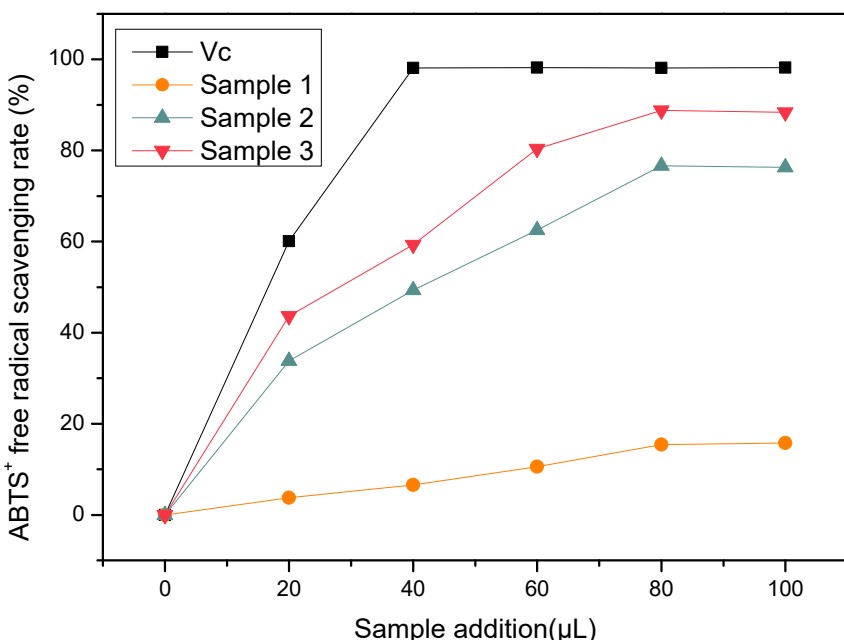

**Figure 7.** ABTS free radical clearance ability.

The clearance rates of sample 1, sample 2, and sample 3 increased gradually with the increase in sample dosage, and reached the maximum values at 80 μL; the ABTS radical scavenging activities were 15.8%, 76.32%, and 88.36%, respectively. The results showed that the fruit had a greater ability to scavenge free radicals and reduce oxidative stress [39,40]. However, after fermentation, the ABTS+ clearance rate of elderberry wine increased, indicating that fermentation made the elderberry release some antioxidant substances and produced beneficial effects [41].

### 3.3.3. Superoxide Anion $O_2^-$ Radical Scavenging Ability

From Figure 8, it can be seen that the $O_2^-$ free radical scavenging rate of the 3 samples increased as the number of samples added increased. When the dosage was 80 μL, the $O_2^-$ radical scavenging ability of sample 1 and sample 2 reached their maximum values, which were 42.34% and 65.64%, respectively. When the addition amount was 60 μL, the scavenging ability of sample 3 on $O_2^-$ radicals reached 88.4%. It can be seen that the $O_2^-$ radical scavenging ability of fermented elderberry wine was significantly improved compared with unfermented samples [42].

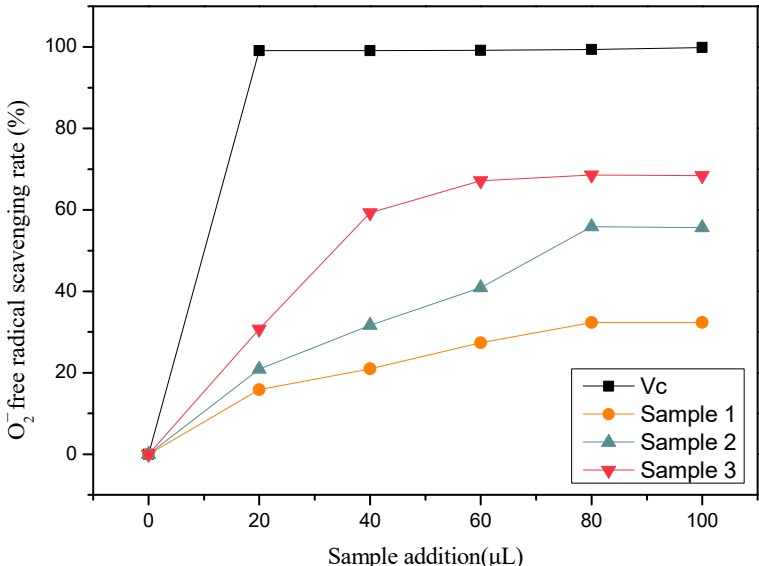

**Figure 8.** The $O_2^-$ free radical clearance ability.

### 4. Conclusions

In this study, a new type of fruit wine was prepared using apples, pears, blueberries, lychees, and elderberries. The response surface method was used to further optimize three factors, based on the results of the single-factor experiment: the solid–liquid ratio (A), yeast inoculum (B), and fermentation temperature (C). The model can better predict and explain the results of the sensory evaluation and alcohol content of fermented elderberry wine. The results of optimization demonstrated that the optimal fermentation conditions were a solid–liquid ratio of 1:7.5, a yeast inoculum of 0.68 g/L, a fermentation temperature of 20 °C, and an added sugar content of 120 g/L. Under these conditions, a 35 L fermenter was selected for the validation test. The final product had a significant influence on the fruity aromas, which contain an apple flavor, pear flavor, and lychee flavor, with a faint flavor of fresh grass, and the aroma is harmonious. The wine was crisp and delicate on the palate, with the sweetness of fruit and the sour taste of blueberries. Following optimization, the volatile components of the samples were analyzed by GC-MS, and the total antioxidant activity was determined. The analysis showed 33 flavor components. The in-vitro antioxidant experiments revealed that its ability to scavenge DPPH, ABTS, and $O_2^-$ radicals was significant. In addition, by comparing the results depending on whether elderberry was involved in fermentation, the results indicated that the antioxidant

capacity of fermented elderberry was improved. The study shows that the optimum process conditions, as determined by fruit concentrate, are feasible, enabling the production of satisfactory and high-quality elderberry fruit wines with a characteristic flavor.

**Author Contributions:** Conceptualization, P.Z.; methodology, H.C. and M.B.; software, H.C.; valida­tion, H.C. and P.Z.; formal analysis, H.C., M.B. and Y.L.; writing—original draft preparation, H.C., M.B., X.Y., C.Z. and K.L.; writing—review and editing, P.Z. All authors have read and agreed to the published version of the manuscript.

**Funding:** This research was funded by "Excellent master thesis training program of Henan Province", School of Life Sciences, Henan University (No. SKYYSPY2022016).

**Institutional Review Board Statement:** Not applicable.

**Informed Consent Statement:** Informed consent was obtained from all subjects involved in the study.

**Data Availability Statement:** Not applicable.

**Conflicts of Interest:** The authors declare no conflict of interest.

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
