# Peer review of "Optimization of the Brewing Process and Analysis of Antioxidant Activity and Flavor of Elderberry Wine"

_fermentation, doi:10.3390/fermentation9030276_

Round 1

Reviewer 1 Report

Fermentation-2248132

Optimization of the Brewing Process and Analysis of Antioxidant Activity and Flavor of Elderberry Wine

The submitted ms is very interesting and provides a thorough investigation of parameters of fruit wines. It is worth of publication provided that the following minor points of analytical interest are taken into consideration:

1.     Table 4 and lines 270 & 271: R2 should always be given with the number of measurements used for calculation.

2.     Table 5: Probably the ±numbers are standard deviations? If yes, this must be made clear on the Table and the number of measurements for each standard deviation should be included.

Author Response

Reviewer:The submitted ms is very interesting and provides a thorough investigation of parameters of fruit wines. It is worth of publication provided that the following minor points of analytical interest are taken into consideration:

  1. Table 4 and lines 270 & 271: R2 should always be given with the number of measurements used for calculation.

Response: Thank you for your suggestion. As what you have suggested, we have made the corresponding changes in the text. It is indicated in Table 2 and line 141.

  1. Table 5: Probably the ±numbers are standard deviations? If yes, this must be made clear on the Table and the number of measurements for each standard deviation should be included.

Response: Thank you for your comments. As the reviewer requested, we have indicated it explicitly in Table 5 and the number of measurements for each standard deviation has been included.

Reviewer 2 Report

This manuscript corresponds to the trends in the production of fruit wines and presents interesting findings in this area. It is especially important for the production of new fruit wine obtained from different fruit juice concentrates.

Figure 1. The results of the single factor test needs to be improved in terms of positioning and size.

From reference 10 onwards, renumbering should be carried out.

Author Response

Author responses

Manuscript ID: Fermentation (ISSN 2311-5637)

Title: Optimization of the Brewing Process and Analysis of Antioxidant Activity and Flavor of Elderberry Wine

Reviewer :This manuscript corresponds to the trends in the production of fruit wines and presents interesting findings in this area. It is especially important for the production of new fruit wine obtained from different fruit juice concentrates.

1.The results of the single factor test needs to be improved in terms of positioning and size.

Response: Thank you for your comments. The positioning and size of the single factor tests have been changed in the text.

2.The positioning and size of the single factor tests have been changed

Response: The reference numbers have been changed.
